# Evaluation of a High-Sensitivity Organ-Targeted PET Camera

**DOI:** 10.3390/s22134678

**Published:** 2022-06-21

**Authors:** Justin Stiles, Brandon Baldassi, Oleksandr Bubon, Harutyun Poladyan, Vivianne Freitas, Anabel Scaranelo, Anna Marie Mulligan, Michael Waterston, Alla Reznik

**Affiliations:** 1Department of Physics, Lakehead University, Thunder Bay, ON P7B 5E1, Canada; jtstiles@lakeheadu.ca (J.S.); obubon@lakeheadu.ca (O.B.); areznik@lakeheadu.ca (A.R.); 2Radialis Medical, Thunder Bay, ON P7A 7T1, Canada; michael.waterston@radialismedical.com; 3Thunder Bay Regional Health Research Institute, Thunder Bay, ON P7B 7A5, Canada; hpoladya@lakeheadu.ca; 4Department of Medical Imaging, University Health Network, Sinai Health System, Women’s College Hospital, Toronto, ON M5S 1B2, Canada; vivianne.freitas@uhn.ca (V.F.); anabel.scaranelo@uhn.ca (A.S.); annamarie.mulligan@uhn.ca (A.M.M.); 5Department of Medical Imaging, University of Toronto, Toronto, ON M5S 1A1, Canada

**Keywords:** organ-targeted PET, breast cancer, precision medicine, cancer detection, detectors, functional imaging, low-dose imaging

## Abstract

The aim of this study is to evaluate the performance of the Radialis organ-targeted positron emission tomography (PET) Camera with standardized tests and through assessment of clinical-imaging results. Sensitivity, count-rate performance, and spatial resolution were evaluated according to the National Electrical Manufacturers Association (NEMA) NU-4 standards, with necessary modifications to accommodate the planar detector design. The detectability of small objects was shown with micro hotspot phantom images. The clinical performance of the camera was also demonstrated through breast cancer images acquired with varying injected doses of 2-[fluorine-18]-fluoro-2-deoxy-D-glucose (^18^F-FDG) and qualitatively compared with sample digital full-field mammography, magnetic resonance imaging (MRI), and whole-body (WB) PET images. Micro hotspot phantom sources were visualized down to 1.35 mm-diameter rods. Spatial resolution was calculated to be 2.3 ± 0.1 mm for the in-plane resolution and 6.8 ± 0.1 mm for the cross-plane resolution using maximum likelihood expectation maximization (MLEM) reconstruction. The system peak noise equivalent count rate was 17.8 kcps at a ^18^F-FDG concentration of 10.5 kBq/mL. System scatter fraction was 24%. The overall efficiency at the peak noise equivalent count rate was 5400 cps/MBq. The maximum axial sensitivity achieved was 3.5%, with an average system sensitivity of 2.4%. Selected results from clinical trials demonstrate capability of imaging lesions at the chest wall and identifying false-negative X-ray findings and false-positive MRI findings, even at up to a 10-fold dose reduction in comparison with standard ^18^F-FDG doses (i.e., at 37 MBq or 1 mCi). The evaluation of the organ-targeted Radialis PET Camera indicates that it is a promising technology for high-image-quality, low-dose PET imaging. High-efficiency radiotracer detection also opens an opportunity to reduce administered doses of radiopharmaceuticals and, therefore, patient exposure to radiation.

## 1. Introduction

The emergence of new radiotracers for positron emission tomography (PET) is continuing to expand its impact on clinical practice. The development of new precision radiotracers binds imaging activity to specific clinical targets, advancing personalized (or precision) medicine [1,2,3,4]. In addition to scanning of the body with sequentially performed whole-body (WB) PET/CT (computed tomography) scanners and emerging simultaneous PET/MRI (magnetic resonance imaging), the applications for PET imaging increasingly involve the visualization of specific organs with dedicated systems [5,6,7,8]. Compared to WB PET scanners, an organ-targeted PET system is capable of higher efficiency, higher spatial resolution, and higher signal-to-noise ratio, resulting in better image contrast and enabling more precise PET examinations. Indeed, an organ-targeted PET camera with optimized geometry can position detectors in close proximity to the organ of interest to facilitate (1) more efficient gamma-ray detection; (2) higher spatial resolution; and (3) reduced unwanted signal from elsewhere in the body, improving the noise-equivalent count rate (NECR) within the field of view (FOV) due to a reduction of false coincidences [7,8,9,10]. Potentially, this may significantly lower a radiotracer dose, thereby reducing radiation exposure associated with PET molecular imaging.

Currently, the average standard-of-care dose of 2-[fluorine-18]-fluoro-2-deoxy-D-glucose (^18^F-FDG) radiotracer for both WB PET/CT and organ-targeted PET examinations is 380 MBq, which gives an effective dose of ~5 mSv. Even though organ-targeted PET significantly reduces the effective dose compared to PET/CT (due to the elimination of the CT component, which accounts for 80% of patient dose during PET/CT), PET radiotracers still deliver a dose ~10 times more effective than, for example, X-ray mammography [11,12]. Therefore, PET detectors must be capable of imaging at significantly lower doses for use in one of the most clinically demanding applications for breast imaging in women with dense breasts (for whom the chance of inaccurate imaging findings is increased due to low sensitivity of X-ray mammography in dense breast tissue). Ideally, a 10-times dose reduction is needed; however, even the capability of producing high-contrast images at ~1/5 of a standard ^18^F-FDG dose will significantly expand patient populations for whom PET is an appropriate imaging modality. This includes women at high risk of breast cancer development, pediatric patients, and patients who require multiple nuclear medicine examinations such as evaluation of treatment response, etc. 

Overall, although organ-targeted PET has the potential for new advances in diagnosis and theranostic procedures, from cancer to cardiac and neuroimaging [13], a significant improvement in PET detector sensitivity is required so that the relatively high whole-body radiation exposure is resolved [14,15,16,17]. In addition, improved sensitivity may shorten exam times, improving patient comfort, minimizing motion artifacts, and increasing patient throughput. 

Another constraint that limits widespread adoption of organ-targeted PET in clinical practice is organ-specific geometries for use with very specific indications. A versatile design that resolves geometric constraints of existing organ-targeted PET technologies may extend use beyond a single target organ, which can permit higher rates of utilization.

The above problems are addressed in this publication: We report on the Radialis PET Camera [18,19,20], which is a versatile, high-sensitivity solid-state PET camera developed for low-dose organ-targeted imaging. A clinical prototype of the Radialis PET Camera evaluated here is optimized for imaging in breast; however, its modular technology offers a flexible geometry that may be adapted for other possible indications [14].

The Radialis PET Camera is evaluated in terms of the activity sensitivity, system count rates, and spatial resolution, and is compared to other commercially available systems. In addition, selected results from a clinical study in progress to evaluate the performance of the Radialis PET Camera with low radiotracer (i.e., 2-[fluorine-18]-fluoro-2-deoxy-D-glucose (^18^F-FDG)) activity are presented. 

## 2. Materials and Methods

The Radialis PET Camera employs two planar detector heads mounted on a movable gantry (Figure 1). Each detector head contains 12 four-side tileable (mosaic) sensor modules that are arranged against each other in a 3 × 4 array (Figure 2A) to assemble a uniform planar-sensing area. Each sensor module (Figure 2B) contains a Cerium-doped Lutetium Yttrium Orthosilicate (LYSO) scintillation crystal (pixelated to make a 24 × 24 grid with an individual pixel size of 2.32 mm × 2.32 mm × 13 mm) coupled to an array of 8 × 8 silicon photomultipliers (SiPMs) of the Array-C type developed by ON Semiconductor (Phoenix, AZ, USA): LYSO crystal array converts gamma photons into visible light while a SiPM tile converts visible light into a readout signal. The LYSO crystals and SiPMs are optically coupled through uncoated 5 mm-thick borosilicate light guides, which allow light sharing over multiple SiPM pixels for the use of light-sharing coordinate reconstruction methods. The light guide is tapered such that the top and bottom face match the dimensions of the scintillating crystal and the photodetector respectively [18], to allow a geometric match between the size of the LYSO crystal and the SiPM. As can be seen from Figure 3, all the modules’ components and front-end electronics are configured such that none of the components are larger than the scintillating crystal, allowing the sensor modules to be seamlessly combined into a 3 × 4 array.

The size of the individual sensor modules shown in Figure 2B is 57.66 mm × 57.66 mm, which results in a seamless sensor area of 230.64 mm × 172.98 mm. The detector housing is made from a thin, durable material so that the imaging area is only ~4 mm from the edge of the detector housing. 

For breast imaging, patients are seated upright with the detector heads positioned on either side of the immobilized breast. A gantry with a rotation axis allows for 90-degree rotation of the detector heads clockwise and counter-clockwise from its starting position. This permits the acquisition of breast and axilla images at standard views (i.e., bilateral craniocaudal (CC) and mediolateral oblique (MLO) views) as well as at supplementary views if additional information is required. It also allows a technician to adjust the position and distance between the detector heads to accommodate the patient’s height and breast size. 

The detector architecture relies on multiplexed readout: 64 channels of the SiPM pixels in each sensor module are multiplexed to 4 readout signals from the pre-amplifiers (AB4T-ARRAY64P, AiT Instruments, Newport News, VA, USA). Event coordinates are reconstructed from this 4-channel signal readout, which is applied to an Anger Logic [21] to determine the coordinate of each detected event. Lines of response (LORs) are collected and stored in list-mode format for the reconstruction of an image of the radiotracer distribution by an iterative maximum likelihood expectation maximization (MLEM) method [22]. A median root prior filter [23] is applied within the MLEM reconstruction after each iteration.

The timing and energy windows for image acquisitions are set at 4 ns and 350–700 keV, respectively, and are consistent through each acquisition. 

In addition to the energy-window filter, a LOR angle-allowance filter is implemented to reject events within the list-mode data based on the endpoints of each LOR. If the difference between the coordinates of detected coincident events in the XY plane is larger than a predefined threshold, the event is discarded from further processing. This discriminates oblique LORs and thus reduces the contribution of the parallax effect.

Thermal stability of the SiPMs is achieved by actively cooling the detector arrays using the built-in temperature-control unit and maintaining the operating temperature of the detector head at 18 ± 1 °C. This cooling approach allows for the stable operation of the detector heads during image acquisition in a clinical setting. 

Since there is no National Electrical Manufacturers Association (NEMA) standard designed for organ-targeted limited FOV PET systems, the performance of Radialis PET was evaluated with the NEMA NU-4 2008 standards [24] for pre-clinical scanners. Indeed, the NEMA NU-4 2008 standards designed for small-size ring detectors are more appropriate for the Radialis PET Camera than the NEMA NU-2 standards for WB PET/CT (computed tomography) scanners (the FOV of organ-targeted PET cannot accommodate the large phantoms required for NEMA NU-2).

Tests of detector performance are conducted with the aim of determining spatial resolution; total, true, scattered, random, and noise-equivalent count rates; evaluation of system sensitivity; and evaluation of image quality.

It should be noted that the coordinate system defined by the NEMA NU-4 protocol assumes a ring geometry of small-animal PET scanners and refers to axial or trans-axial directions for measurements of spatial resolution. However, the planar geometry of the Radialis PET Camera is described in the cartesian coordinates shown in Figure 4 with the XY plane parallel to the detector heads and the Z-axis, which points from one detector head to the other. A single-slice re-binning (SSRB) reconstruction method [25,26] was used to process the corrected list-mode data according to Section 4 and Section 5 of the NEMA NU-4-2008 standards. This method assigns each line of response to an image plane halfway between the detector heads based on the intersection of the LOR with the plane. In addition, the NEMA NU-4-2008 standard mandates the derivation of spatial resolution through the reconstruction of point-source images using a filtered back-projection (FBP) technique. Since clinical and phantom images are reconstructed using MLEM algorithm, acquisitions for spatial resolution measurements are reconstructed and compared using both back-projection and iterative MLEM algorithms. 

### 2.1. Spatial Resolution

Spatial resolution was measured by imaging a point source (0.3 mm-diameter Na-22 source encased within an acrylic cube with dimensions of 10 mm × 10 mm × 10 mm) across the FOV. The original activity of the Na-22 source was 111 kBq (3 µCi) and the calibrated activity of the source during these experiments was determined to be 89.9 kBq. Profiles of each source were created in ImageJ [27] by plotting through the maximum intensity pixel of the source and measuring the image gray value along that line. An image pixel size of 0.2 mm for the XY plane and a voxel depth of 2.67 mm was used for the image matrix size throughout the spatial resolution analysis.

Two reconstruction methods were implemented for the analysis of spatial resolution: both the standard MLEM reconstruction with 15 iterations, and a back-projection reconstruction. The system was calibrated using a flood-scan uniformity acquisition. No other corrections were applied to these data sets, such as scatter, attenuation, or dead-time corrections. Spatial resolution was reported in terms of full width at half-maximum (FWHM) and full width at tenth maximum (FWTM) of the point-spread function (PSF), which were determined from a Gaussian fit of the data distribution. Here, we followed a widely adopted practice for PSF characterization [28,29,30], although this slightly deviated from NEMA NU-4 requirements that derives FWHM from line profiles drawn through the image of the point source. It is also noted that no background activity was included with the point-source acquisitions.

The separation between the detector heads was kept at 80 mm for each point-source acquisition. First, spatial resolution was evaluated at the center of the XY FOV half the distance between the detector heads (i.e., at x = 0 and y = 0). Then, the resolution was measured as a function of the distance from the center of the XY FOV along the X- and Y-axes at a Z-location of one quarter the detector-head separation. Each measurement was calibrated to run until more than 100,000 prompt counts were acquired. Resolution was quoted for each axial direction as either X-, Y-, or Z-resolution corresponding to the direction of the profile across the image.

Additionally, the micro hotspot phantom was used for the qualitative assessment of system resolution through the visualization of its small rods. The phantom was filled with 1 MBq of ^18^F-FDG and acquired for 40 min with a detector-head separation of 89 mm, which was dictated by the phantom size. The phantom was immobilized by the detector heads and placed centrally in the X- and Y-directions. For image reconstruction, a pixel size of 0.2 mm × 0.2 mm was used to allow for the visualization of smaller details. Post-processing of the micro hotspot phantom was implemented in ImageJ with a 3D Gaussian blur (sigma = 1.0 pixels) and with an unsharp mask (sigma = 7.0, mask weight = 0.6).

### 2.2. Sensitivity

The same Na-22 point source from the spatial resolution analysis was used for evaluating sensitivity. The source was positioned in the center of the XY plane, halfway between the detector heads in the Z-axis, and moved across the full FOV with 2 mm steps along the X-axis. This results in 113 discrete positions for which sensitivity was calculated. At each position, an image of the Na-22 point source was acquired; the acquisition was calibrated to acquire list mode data for 60 s, yielding enough events for the analysis. Detector heads were separated by 60 mm during each acquisition. Each data set was reconstructed with an SSRB image-processing algorithm. Values of the axial sensitivity S_i_ (Equation (1)) and the absolute per-slice slice sensitivity S_A,i_ (Equation (2)) [24] were determined and plotted as a function of source location in the FOV.
S_i = ((R_i−R_(B,i))/A_Cal)(1)
S_(A,i) = (S_i/0.9060) × 100(2)
where R_i_ is the count rate measured for slice i, R_B,i_ is the background count rate for slice i, and A_Cal_ is the calibrated activity of the source. Absolute sensitivity was calculated with the branching ratio of Na-22 (i.e., 0.9060) and the calculated sensitivity S_i_ for slice i. The average system sensitivity was determined by summing the axial sensitivity across the FOV and dividing by the number of datapoints.

### 2.3. Count-Rate Performance

A NEMA NU-4 (rat) scatter phantom was used for the determination of count-rate statistics. The phantom consists of a long, cylindrical high-density polyethylene (0.98 g/cm^3^) with a diameter of 50 mm and a length of 150 mm. The line source consists of a cylindrical cavity with a diameter of 3.5 mm drilled lengthwise through the phantom at an axial offset of 17.5 mm and filled with 51 MBq of F-18 solution. The line source is closed at each end with 4 mm-long syringe ports, resulting in an overall length of 142 mm.

The phantom was placed at the center of the XY FOV (y = 0) parallel to the X-axis halfway between the detector heads in Z with a separation of 60 mm. Acquisitions began immediately after the phantom was filled and were programmed to repeat every 15 min until the phantom had decayed through 10 half-lives and a maximum of 29 million total events had been acquired. Negligible amounts of activity remained in the final acquisitions.

Data processing for count rates involved reconstructing list-mode acquisition data files using LOR acceptance-angle filtration (i.e., only LORs whose endpoints have certain DX and DY, referred to as “angle allowance,” were used for image processing; Figure 4). The resulting files were then processed using an SSRB image-reconstruction technique. Peak count rates were determined from the plots of count rates vs. phantom-activity concentration.

NECR performance was evaluated over a clinically relevant activity range and efficiency at peak noise equivalent count rate was determined as the peak NECR normalized to the activity at the peak:Eff_(NECR,peak) = NECR_(Peak)/A_(Peak)(3)

### 2.4. Clinical Imaging

The Radialis PET Camera was tested, and it is currently in use for a clinical trial [31] at the Princess Margaret Cancer Centre of the University Health Network (UNH-PMCC) in Toronto, Canada. Participants in the study received a clinical indication for diagnostic medical imaging tests such as full-field digital mammography (FFDM) with or without digital breast tomosynthesis (DBT), breast MRI, or WB PET/CT scan.

Patients with a newly diagnosed breast cancer were injected with ^18^F-FDG in the range of activities between 37 and 307 MBq (activity was chosen randomly and did not depend on the clinical case). Each participant rested for 60 min to allow for the ^18^F-FDG uptake. Some participants who received WB PET/CT were first imaged with the Siemens Biograph Vision WB PET/CT scanner (Erlangen, Germany) (image acquisition time ~30 min), and immediately after that they were taken for another imaging session with the Radialis system (single image-acquisition time ~5 min at each position). Obtaining WB-PET/CT and Radialis PET images permitted a direct comparison between the two PET-imaging modes. For some patients for whom WB PET/CT was not indicated, they were imaged with the Radialis PET system and the breast-imaging modalities (i.e., FFDM, FFDM-DBT, MRI) alone or in combination. Optionally, a second set of images was acquired for patients who received 185 MBq of ^18^F-FDG and opted to return in two hours for a subsequent imaging session when the ^18^F activity had decayed to approximately 1/4 of the initial activity (~4 h post-injection).

## 3. Results

### 3.1. Spatial Resolution

Results of the average spatial resolution as a function of point-source location along the Y-axis and X-axis are presented in Table 1 and in Figure 5, showing the X, Y, and Z MLEM resolutions as functions of location. The values of the PSF full width at tenth maximum (FWTM) are also provided, in addition to the spatial resolution values in terms of FWHM. In-plane spatial resolution, determined by the X- and Y-resolution plots, had an average value of 2.3 ± 0.1 mm. The resolution for the system along the Y-direction stayed consistent across the entire FOV, with an average value of 2.3 ± 0.1 mm. Similarly, the resolution along the X-direction of the system maintained a FWHM of 2.2 ± 0.1 mm. As expected, the cross-plane or Z-resolution of the system was about three times larger than in-plane resolution and had an average value of 6.8 ± 0.7 mm within the central FOV.

The results from the same acquisition reconstructed with a back-projection algorithm are presented in Figure 6. The in-plane resolution for the central Z-axis location was, on average, 3.3 ± 0.1 mm and the cross-plane resolution was 16.4 ± 0.1 mm.

The reconstructed image of the micro hotspot phantom presented in Figure 7 demonstrates the visualization of small sources down to the 1.35 mm-diameter rods while using the MLEM reconstruction and down to the 1.7 mm-diameter rods with the back-projection reconstruction. As expected, MLEM-reconstructed images exhibited less noise and better image contrast.

### 3.2. Sensitivity

Sensitivity values for the system are displayed as functions of point-source location along the X-axis in Figure 8 and are summarized in Table 2. A peak axial sensitivity value of 32 cps/kBq is shown at the center of the FOV, which, after normalizing to the branching ratio of Na-22, gave a peak absolute axial sensitivity of 3.5%. Figure 8 demonstrates the details of the measured NEMA sensitivity profile along X-axis positions: As expected, sensitivity gradually decreased as the source was moved towards the edge of the detector head since the LOR’s solid angle decreased. Total average system sensitivity is determined as the sum of each single value along the plot divided by the number of datapoints and was equal to 2.4%.

### 3.3. Count-Rate Performance

Count rates for the system were plotted against the scatter phantom activity concentration and included the prompt, true, scatter, noise equivalent, and random count rates in Figure 9. Activity concentrations corresponding to specific standard uptake values (SUV) were marked on each count-rate plot (SUVs were calculated for different clinically relevant injected activities for a 77.3 kg woman). The values of SUV = 1 were included to estimate the activity that would be expected for the background tissue during acquisition. Peak count rates are summarized in Table 3 for several different LOR angle-allowance filters. As is evident from Figure 9, peak noise equivalent count rates (NECRs) were achieved at a phantom activity concentration of 10.5 kBq/mL with an efficiency at peak NECR of 5650 cps/MBq. The scatter fraction for the 90 mm and 110 mm LOR angular filters were 24% and 31% respectively. The use of a 52 mm LOR angle allowance filter further reduced the scatter fraction to 6.2% accompanied by a cut to the overall count rates.

### 3.4. Clinical Imaging

Figure 10 compares the FFDM craniocaudal (CC) view (Figure 10A) with a low-dose Radialis PET image in a CC view (Figure 10B) acquired in a 56-year-old female with histopathology-diagnosed invasive ductal carcinoma and intermediate-grade ductal carcinoma in situ (DCIS). For the PET imaging, the patient received intravascular 37 MBq of 18F-FDG, and the scanning was performed 1 h after injection. The focal uptakes on the Radialis PET image (arrow and arrowhead in Figure 10B) corresponded to one mass (arrow in Figure 10A) detected in FFDM; however, the other mass that was also histopathology proven was detected only in the Radialis PET images. The second cancer was not detected by mammography, even in retrospect, because of the dense breast tissue-masking effect.

Figure 11 shows the comparison among multimodality images, specifically an FFDM CC view (Figure 11A), MRI axial subtracted view (Figure 11B), and two Radialis PET camera CC view images (Figure 11C,D) obtained from a 61-year-old woman with a known malignant disease involving the lateral aspect of the right breast. For the organ-targeted PET acquisition, 178 MBq of ^18^F-FDG was administrated and two subsequent imaging sessions were acquired at 1 h (Figure 11C) and 4 h (Figure 11D) post injection. The PET images showed that changes in image contrast with time as activity decreased were not impactful for the radiologist’s visual assessment of multifocal cancers. Both Radialis PET images demonstrated ^18^F-FDG uptake in the extensive area that corresponds to the irregular mass detected on digital mammography and to a single irregular shape mass demonstrated by MRI images. However, the Radialis PET images were more reproducible of histopathology findings with multiple foci of cancers. Even after 4 h, the PET image (Figure 11D) still showed multiple distinct regions of increased uptake spanning an area of contiguous contrast enhancement on MRI or distortion on FFDM images. 

Figure 12 shows the results of FFDM and Radialis PET Camera imaging in a 50-year-old female with a palpable breast lump against the chest wall. The mediolateral oblique (MLO) digital mammography image identified a single palpable mass. Radialis PET Camera images were acquired with 200 MBq of injected activity, revealing two additional regions of enhanced contrast along the patient’s chest wall, which surgical pathology confirmed as malignancy.

Figure 13 presents MRI 3D maximum-intensity-projection images (Figure 13A) showing multiple rounded and oval shape-enhancing masses in both breasts. There was a noticeable discrepancy in MRI-depicted lesions with the lack of focal uptake of 18F-FDG in the Radialis PET images acquired with a 37 MBq injection. This patient underwent programmed bilateral breast surgery (mastectomy) without malignancy identified in the surgical pathology report.

The clinical WB PET images presented in Figure 14 were acquired with a Siemens Biograph Vision WB PET/CT. Figure 14A shows the full FOV slice with the region of the image with the breast expanded in (B) and the Radialis PET Camera (C) of a 50-year-old patient with a known malignancy in the right breast. A dose of 307 MBq of ^18^F-FDG was administered and the WB PET/CT image acquisition was performed after a 60 min uptake time. Immediately after the WB PET/CT examination, the patient was imaged with the Radialis PET Camera. The WB PET/CT axial images identified an inhomogeneous hypermetabolic mass and a slightly hypermetabolic satellite nodule. Despite the shorter imaging time for the Radialis PET acquisition (5 min), the extent of the lesions was more clearly defined, both in terms of the extent of the lesions and the regions within the lesion with the highest functional activity. Smaller anatomical features such as the nipple were visible in the organ-targeted image but were not present in the WB images.

## 4. Discussion

Despite the proven value of using radiotracers in a broad spectrum of diagnostic procedures across oncology, cardiology, and neurology, the standard radiotracer doses used in PET diagnostic procedures of 185 to 370 MBq [32] continues to define the limitations of PET imaging for use in undiagnosed patients (including screening procedures) and radiation-sensitive population [33]. In this regard, much attention is currently being paid in the literature to synthesizing high-quality PET images from input images acquired with a low dose of radiotracers through the use of deep learning and convolutional neural networks (CNNs) [16,34,35]. Despite the high potential of deep learning to denoise low-count PET images, a comprehensive approach is needed so that advances in software are complemented by improved PET-detector sensitivity to make low-dose PET imaging a clinical reality. Improved detector sensitivity will also allow for a reduction in the scanning time required to both minimize the risk of motion artifacts and improve patient throughput—an important element for making PET procedures cost effective [36].

The development of the organ-targeted Radialis PET technology described herein focused precisely on the matter of hardware-based improvements in sensitivity and NECR performance across a clinically useful activity range, down to low-dose activities at 1/10th of a standard dose [37]. This was achieved primarily through detector design and increased geometric coverage and therefore decay count registration by arranging the large FOV detectors proximal to the organ of interest. For imaging in breast, the FOV was made similar to that of mammography, i.e., ~230 mm × 172 mm.

The first clinical evaluation of the developed organ-targeted PET camera was devoted to breast cancer due to the clinical significance of high-sensitivity molecular breast imaging with ^18^F-FDG PET: It has the potential to overcome a well-known drawback of mammography—low sensitivity in heterogeneous and extremely dense breasts [38], found in roughly 50% of the population [39]. Since breast ^18^F-FDG PET uptake is largely independent of breast-tissue density, it can overcome the lesion obscurity (masking effect) experienced in mammography from dense breast tissue. We also demonstrate a potential to address the high false-positive rate associated with gadolinium-enhanced breast MRI, shown with clinical images.

Several PET systems have been developed for imaging clinically relevant breast cancers with performance independent of breast density and hormonal changes [38,40,41]. Breast-targeted PET systems differ from WB PET detectors in both geometry (using either planar or ring detectors) and positioning of the breast during image acquisitions. The Naviscan Positron Emission Mammography (PEM) Flex Solo II uses two planar compression heads positioned on either side of a breast, containing line detectors that scan across the FOV. The reported in-plane spatial resolution for the Flex Solo II PEM using MLEM reconstruction is 2.4 ± 0.2 mm [22]—a significant improvement over WB PET (which is 5–7 mm [42,43]), and cross-plane resolution is 8.2 ± 1.0 mm [44]. However, since the instantaneous coverage of the moving detectors is a fraction of the full FOV, this method results in longer acquisition times, decreased peak slice sensitivity and peak slice absolute sensitivity (0.2% and 1.8 cps/kBq), and higher effective dose exposures (370 MBq injection [45]). Despite this, the Naviscan system has a higher sensitivity than MRI for the smallest cancers (in part since it is not angiogenesis dependent) [29,46].

A more recent organ-targeted PET technology is Mammi Breast PET, developed by Oncovision, which uses a circular array of 12 detectors (or two circular arrays in a high-sensitivity configuration). For imaging with this system, the patient lies prone and the breast hangs pendulant into the detector ring. Although this design allows for greater sensitivity and faster scanning times, it has reduced imaging capabilities for lesions near the chest wall [47]. The peak per-slice absolute sensitivity of this system is improved compared to the Flex Solo II system, with 1.8% for the single-ring configuration and 3.1% with the dual-ring configuration, which offers thicker scintillation crystals and a larger detection area [48,49]. The ring configuration of the MAMMI PET system is able to achieve nominally higher point-source resolution [49] than the Radialis PET Camera (1.5–1.9 mm vs. 2.2–2.4 mm). However, the sensitivity for identifying clinically relevant cancers also depends on the overall efficiency of activity detection, and the peak slice sensitivity and peak absolute slice sensitivity of the Radialis PET Camera is larger than in both the Flex Solo II and Mammi Breast PET systems.

It should be noted that the NEMA-NU2 method for measurement of WB PET system sensitivity uses a line source, whereas the NEMA-NU4 method adapted here for organ-targeted systems uses a point source, which makes a significant difference in the meaning of the sensitivity and its values. Therefore, measuring NECR—another parameter that characterizes the efficiency of activity detection in PET imaging—for the entire activity of the extended source is necessary for performance comparison between different PET technologies. NECR describes the true coincidence rate that would give the observed signal-to-noise ratio (SNR), or the same level of statistical noise, if there were no random or scattered events detected. Table 4 presents the efficiency at peak count rate for several PET systems, including organ-targeted, whole-body, and emerging total-body systems. The Radialis PET Camera exhibits much higher efficiency at peak count rate when compared to current WB systems. The SiPM-based total-body PET technology of uExplorer [50] and Biograph Vision Quadra [51] provides superior sensitivity in comparison to the WB systems, achieved with detectors that can completely cover the axial length of a patient’s body. Radialis’ SiPM-based organ-targeted technology applies the same approach: The geometric coverage of the Radialis SiPM arrays is larger than the organ being imaged. The increased axial extent of the detectors and the absence of dead zones between sensor modules provide more efficient detection of annihilation events than in other systems dedicated to imaging in breast (i.e., Oncovision Mammi PEM and Naviscan PEM Flex Solo II).

Optimized for low-dose imaging, the count rate for the Radialis PET Camera peaks at relatively low activity values. However, Figure 9 illustrates that the coincidence count-rate capabilities and the dead-time characteristics are still favorable for standard clinical doses. The equivalent SUV values are indicated for a standard clinical range of injected activity from 185 to 370 MBq (5 to 10 mCi), as well as low-dose 37 MBq (1 mCi) imaging. For SUV 1-7 at 370 MBq the count rates are no worse than 78% of the peak NECR. It is presumed possible that administered activity may be reduced without significant compromise in imaging results by way of the higher sensitivity and low activity count-rate peaks. 

Current clinical practice for breast imaging with a dedicated PET system requires an injection of 370 MBq (10 mCi) of ^18^F-FDG [31,40], resulting in an effective dose to the breast of 3.4 mGy and an effective whole-body dose of up to 6.2–7.1 mSv [15,45]. This effective dose is more than 10 times the average effective dose of 0.5 mSv for digital mammography [15,45]. From the standpoint of radiation-induced cancer risks, the injected ^18^F-FDG activities need to be reduced to 70 MBq or less [45] for PET to be considered as an alternative to DBT or breast MRI for screening of high-risk women. In this case an effective radiation dose of ~1.3 mSv is estimated to be equivalent to the effective dose from combined FFDM with DBT [15].

Figure 10 illustrates the capability of the Radialis PET Camera to image with 37 MBq of activity (10 times lower than the standard dose), suggesting that further study of the clinical sensitivity for breast cancer detection with 70 MBq of radiotracer is warranted.

Images taken after two different time intervals (Figure 11) demonstrate the image quality at a reduced count rate due to radiotracer decay as well as increased lesion-to-background ratio over time due to the different wash-out mechanism for cancerous and background tissue [56]. With the camera’s high sensitivity for low count-rate acquisitions, the images present a stronger discrimination of multiple foci over time, even though the activity is reduced through the decay of the injected radiotracer. 

Figure 12 demonstrates the importance of the thin detector heads of the Radialis PET Camera and the small distance to the front of the field of view for improving the visualization of deep chest lesions—a recognized challenge for breast-specific PET systems [8,9]. This is an important differentiation from pendulant breast PET systems, where lesions residing near the chest wall are outside of the field of view and cannot be imaged due to constraints in patient positioning.

The results in Figure 13 show another very important direction in clinical use of the evaluated PET technology, namely, avoiding unnecessary downstream services and overdiagnosis caused by breast MRI [57].

## 5. Conclusions

Along with the scintillator material and photosensor characteristics, the main element that influences PET sensitivity is scanner geometry, which includes the active area of the PET detector intercepting annihilation events [58,59]. Here we show that the sensitivity of organ-targeted PET can be significantly improved with planar-detector geometry provided that the FOV and the distance between two detectors are appropriate for the solid angle available for the collection of annihilation radiation. The optimization of the planar FOV was achieved using tiled block detectors combined with high-yield scintillation crystals, high-gain solid-state photodetectors, temperature control, and acquisition electronics architected for the application. Clinical demonstration with imaging in breast revealed that the Radialis PET technology is well suited to identifying cancers even at a 10-fold dose reduction in comparison with the standard WB PET dose. At a standard dose of ^18^F-FDG, images acquired with the Radialis PET Camera showed clinical detail that cannot be seen with commercial WB PET scanners. 

The demonstrated capability for imaging with less than 70 MBq suggests that Radialis organ-targeted PET technology could be used in clinical applications for undiagnosed patients, including screening procedures. High-quality organ-targeted imaging may also be particularly well suited to applications with emerging targeted radiotracers.

Another important clinical application for organ-targeted PET technology such as the Radialis PET Camera is in addressing emerging clinical demand for assessment of metabolic response in tumors for treatment follow-up. This requires an ability to accurately assess SUV in small lesions and to quantify changes in radiotracer uptake—characteristics that are left for discussion in future publications. 

## Figures and Tables

**Figure 1 sensors-22-04678-f001:**
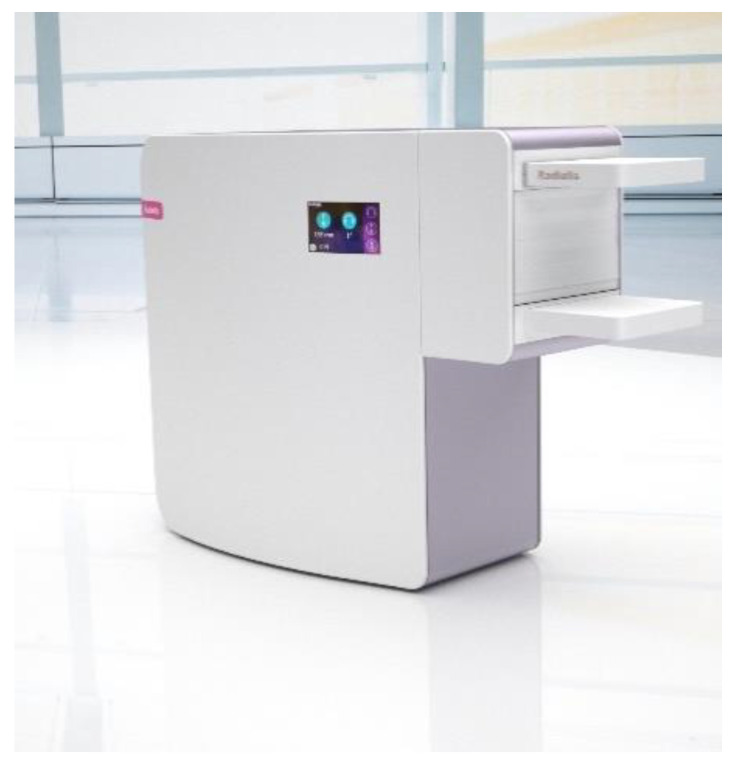
Configuration of the Radialis PET Camera with two planar detector heads to be positioned on either side of a breast.

**Figure 2 sensors-22-04678-f002:**
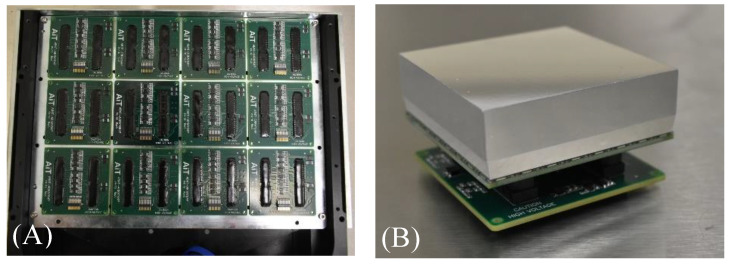
(**A**) Top view of 3 × 4 array of sensor modules inside a detector head; (**B**) photo of a block detector with the crystal array wrapped in a light-reflective material and an electronic board underneath.

**Figure 3 sensors-22-04678-f003:**
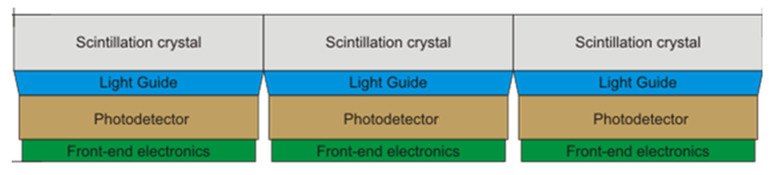
Schematic representation of a cross-sectional view of three tiled block detectors.

**Figure 4 sensors-22-04678-f004:**
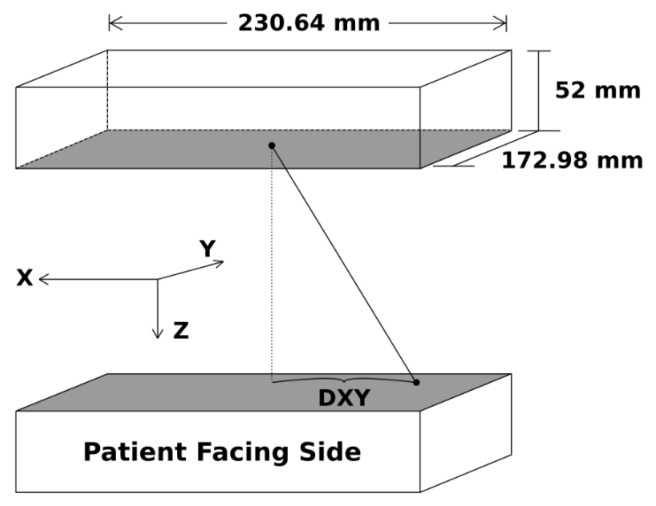
Schematic representation of the overall detector field of view and axis convention.

**Figure 5 sensors-22-04678-f005:**
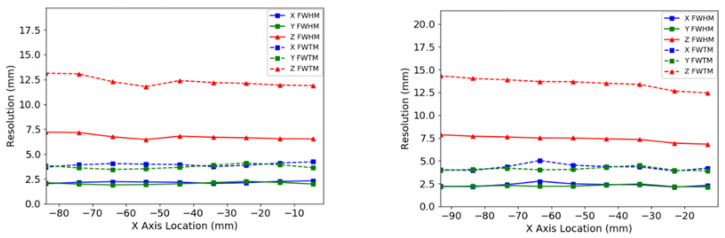
System spatial resolutions produced with the MLEM reconstruction. Left: the central Z-axis resolution plotted as a function of point-source location along the X-axis. Right: the quarter Z-axis resolution plotted as a function of point-source location along the X-axis.

**Figure 6 sensors-22-04678-f006:**
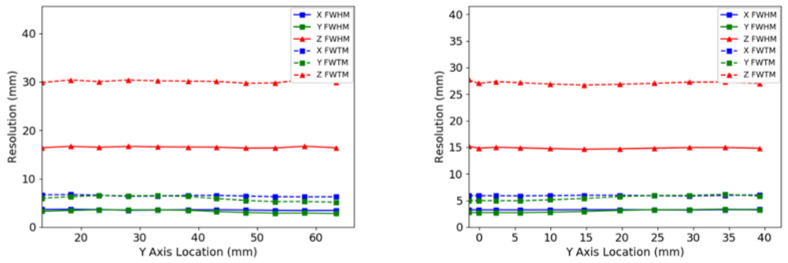
System spatial resolutions produced with a back-projection reconstruction. Left: the central Z-axis resolution plotted as a function of point-source location along the X-axis. Right: the quarter of the Z-axis resolution plotted as a function of point-source location along the X-axis.

**Figure 7 sensors-22-04678-f007:**
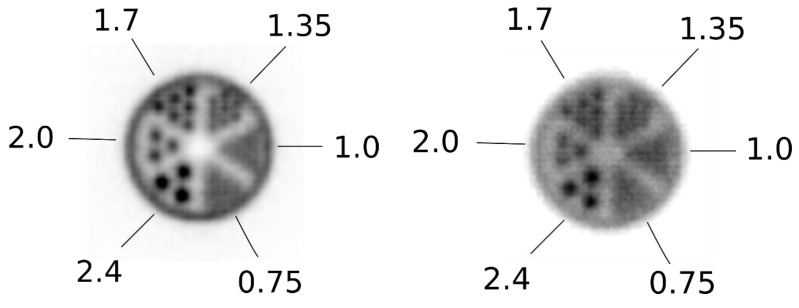
Images of the micro hotspot phantom reconstructed using an MLEM reconstruction (**left**) and with a back-projection reconstruction (**right**).

**Figure 8 sensors-22-04678-f008:**
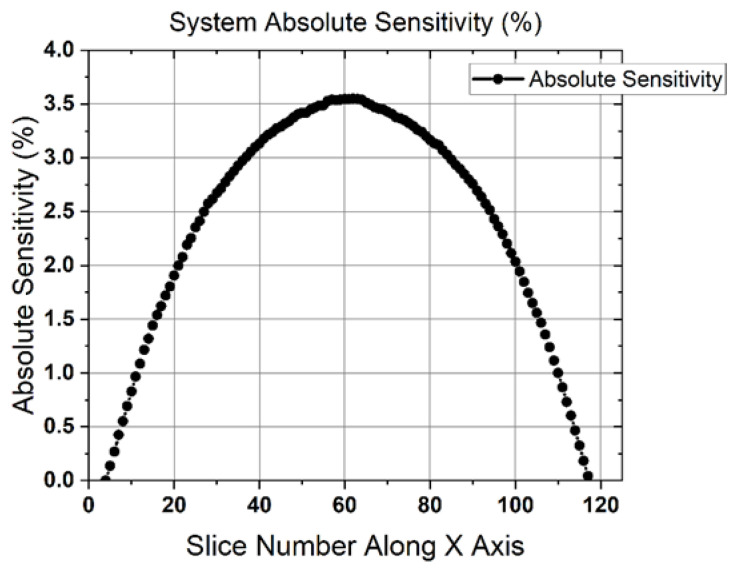
Axial absolute sensitivity plotted against point-source location along the X-axis.

**Figure 9 sensors-22-04678-f009:**
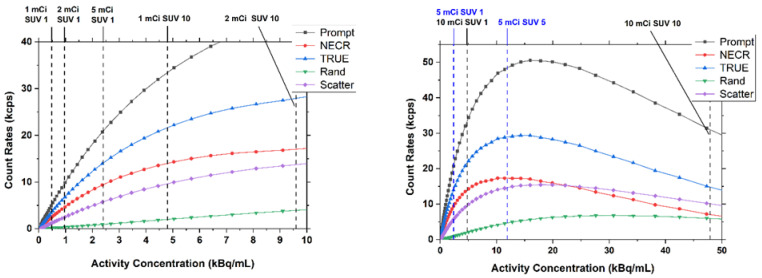
System performance count rates for a 90 mm LOR angle allowance. Left: low range of activity concentrations. Right: full range of activity concentrations.

**Figure 10 sensors-22-04678-f010:**
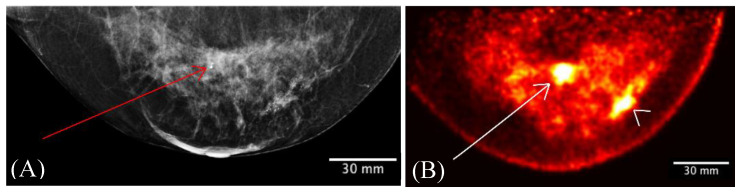
A 56-year-old female with invasive ductal carcinoma and intermediate-grade DCIS. Digital mammography of right breast (**A**) and right breast Radialis PET image with 37 MBq ^18^F-FDG injection (**B**) both in the same projection (CC view) are presented for comparison between these two imaging modalities. Cancers are demonstrated by the arrows (**A**,**B**) and arrowhead (**B**). The second cancer (arrowhead) is visualized only by Radialis PET (**B**).

**Figure 11 sensors-22-04678-f011:**
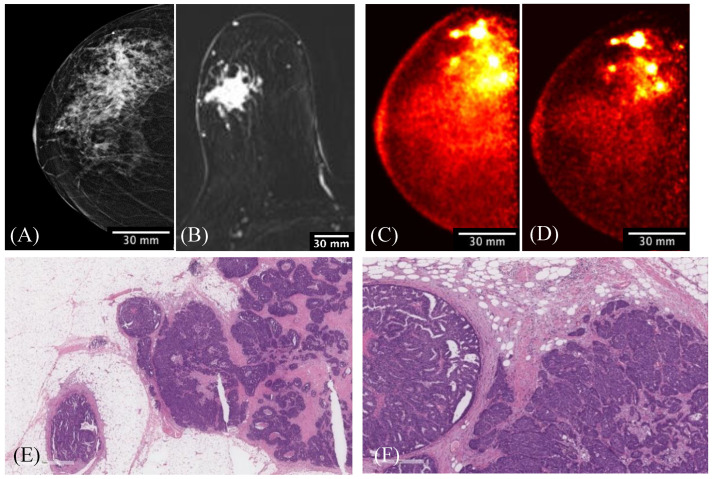
A 61-year-old female with right-breast multifocal invasive and in situ ductal carcinoma. Images of the same breasts in (**A**) FFDM in the CC plane showing extensive distortion, (**B**) a selected slice of MRI in the axial plane showing one irregular shape-enhancing mass lesion after 2 min post gadolinium-chelate-based contrast administration, (**C**) 3D Radialis PET in the CC plane where multiple distinct regions of contrast uptake after 1 h of 178 MBq ^18^F-FDG injection are evident, (**D**) 3D Radialis PET in the CC plane where the conspicuity of the multiple regions of enhanced ^18^F-FDG uptake (indicative of multifocal cancers) remains after 3 h from the prior (**C**) acquisition, (**E**) invasive carcinoma in the center of the field with in situ carcinoma present at the periphery in the pathology of the mastectomy specimen, and (**F**) higher-power view demonstrating intermediate-grade invasive carcinoma on the right and papillary ductal carcinoma in situ on the left.

**Figure 12 sensors-22-04678-f012:**
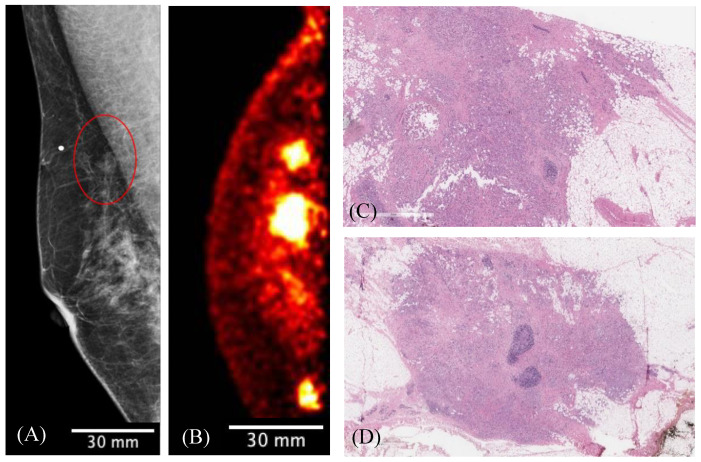
The MLO-view digital mammography image (**A**) demonstrates the palpable mass (red circle) associated with the radiopaque marker placed on the patient’s skin. The presented slice of Radialis PET Camera CC image with 200 MBq injected ^18^F-FDG (**B**) identifies this lesion against the chest wall as well as two additional posterior masses. (**C**) View of the largest focus showing invasive ductal carcinoma of no special type; a clip-site reaction is present in the center of the tumor. (**D**) Second invasive focus demonstrating similar morphologic features and histologic grade. The 3 regions of contrast enhancement identified by Radialis PET are all biopsy-confirmed cancers.

**Figure 13 sensors-22-04678-f013:**
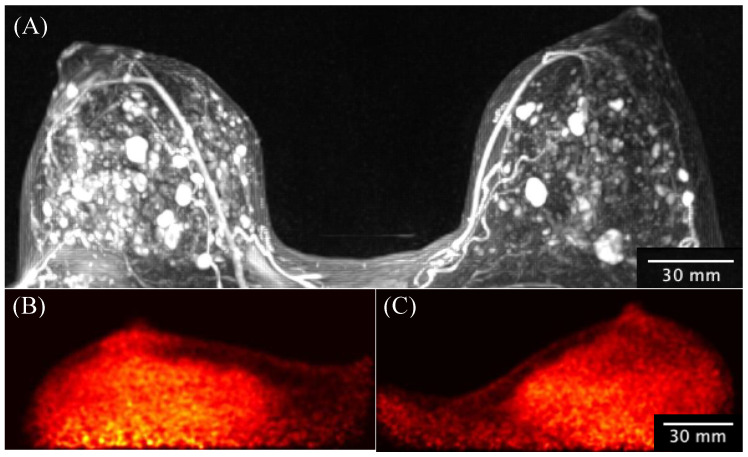
A 33-year-old high-risk female underwent pre-operative breast MRI with multiplicity of enhancing masses demonstrated by the 3D-MIP image (**A**) and without corresponding masses demonstrated by the Radialis PET Camera images (**B**) with a 43 MBq injection. The mediolateral oblique views from the Radialis PET Camera are presented for the left (**B**) and right (**C**) side without evident focal ^18^F-FDG uptake in either image. The surgical pathology results do not show signs of cancer.

**Figure 14 sensors-22-04678-f014:**
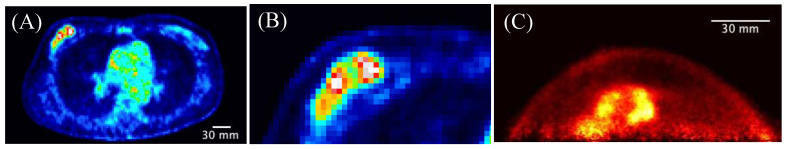
Side-by-side comparison of 307 MBq PET images from a breast cancer patient scanned with a Siemens Biograph PET/CT reconstructed using a time-of-flight reconstruction technique (TOF) (**A**,**B**) and with the Radialis PET system (**C**).

**Table 1 sensors-22-04678-t001:** Average values of MLEM spatial resolution in terms of the X, Y, and Z FWHM and FWTM for a detector-head separation of 80 mm.

	X-Axis		Y-Axis	
**Resolution at Z = 0 mm**	**FWHM**	**FWTM**	**FWHM**	**FWTM**
**In-plane X (mm)**	2.2 ± 0.1	4.1 ± 0.2	2.4 ± 0.2	4.4 ± 0.4
**In-plane Y (mm)**	2.2 ± 0.1	4.1 ± 0.2	2.3 ± 0.1	4.1 ± 0.1
**Cross-plane Z (mm)**	7.8 ± 0.3	14.2 ± 0.5	6.6 ± 0.9	12.1 ± 1.7
**Resolution at Z = 20 mm**	**FWHM**	**FWTM**	**FWHM**	**FWTM**
**In-plane X (mm)**	2.4 ± 0.2	4.3 ± 0.3	2.5 ± 0.1	4.5 ± 0.1
**In-plane Y (mm)**	2.2 ± 0.1	4.1 ± 0.2	2.2 ± 0.2	4.1 ± 0.3
**Cross-plane Z (mm)**	7.3 ± 0.5	13.3 ± 0.9	6.9 ± 0.6	12.7 ± 1.0

**Table 2 sensors-22-04678-t002:** The peak axial absolute sensitivity and the normalized total sensitivity for a 60 mm detector-head separation.

**Detector separation**	60 mm
**Peak absolute slice sensitivity**	3.5%
**Peak slice sensitivity**	32 cps/kbq
**Average total absolute sensitivity**	2.4%

**Table 3 sensors-22-04678-t003:** Summarized values for count rates at different LOR angle-allowance parameters.

	60 mm Angle Allowance (kcps)	90 mm Angle Allowance (kcps)	110 mm Angle Allowance (kcps)	144 mm Angle Allowance (kcps)
**Peak NECR**	9.6	17.3	17.8	18.1
**Peak true rate**	13.8	28.5	32.5	36.4
**Peak prompt rate**	19.7	46.9	59.3	73.1
**Peak scatter rate**	4.15	13.9	20.9	29.3
**Peak random rate**	1.63	4.35	5.76	7.31

**Table 4 sensors-22-04678-t004:** Count rate data for several organ-targeted, whole-body and emerging total-body PET systems.

PET System	Efficiency at Peak Count Rate (cps/MBq)	Peak NECR (kcps)	Concentration at Peak NECR (kBq/mL)	Phantom Volume (mL)	Activity at Peak NECR (MBq)
Radialis PET Camera (NU-4)	5650	17.8	10.5	300	3.15
uExplorer [50] (NU-2) (Total Body)	3790	1440	16.8	22,600	380
Siemens Biograph Vision Quadra (NU-2) [51] (PET/CT)	2666	1613	27.49	22,000	605
Oncovision Mammi PEM Dual Ring (NU-4) [49] (PEM)	1260	34.0	31.2	866	27.0
GE Discovery IQ [42] (PET/CT)	618	123.6	9.1	22,000	200
GE Discovery MI (NU-2) [52] (PET/CT)	581	266	20.8	22,000	458
Phillips Vereos (NU-2) [53] (PET/CT)	556	646	52.8	22,000	1160
GE Signa PET [54] (PET/MR)	524	218	17.8	22,600	402
Siemens Biograph Vision (NU-2) [55] (PET/CT)	435	306	32	22,000	704
Naviscan PEM Flex Solo II (NU-4) [44] (PEM)	393	10.6	90	300	27.0

## Data Availability

Data are available from corresponding author upon reasonable request.

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
