# Peer review of "Evaluation of a High-Sensitivity Organ-Targeted PET Camera"

_sensors, 2022, doi:10.3390/s22134678_

Round 1

Reviewer 1 Report

Stiles and co-workers evaluated the performance of the Radialis organ-targeted positron emission tomography (PET) camera with standardized tests and through assessment of clinical imaging results. High-efficiency radiotracer detection also opens an opportunity to reduce administered doses of radiopharmaceuticals and, therefore, patient exposure to radiation. The results are sound and well supported by their experiments and the manuscript is well-written. The reviewer recommends it for publication after major revision.

1)     The author should further emphasize the significance of high-sensitivity organ-targeted PET camera in the introduction.

2)     Theoretical calculations should be introduced to evaluate the performance of high-sensitivity organ-targeted PET cameras.

3)     The quality of the images in the paper should be improved (e.g., Figure 1-2).

4)     The ruler is missing in many pictures (e.g., Figure 12-14).

5)     Some relevant literature on tumor imaging should be cited (e.g., Coordination Chemistry Reviews, 2021, 430: 213662).

Reviewer 2 Report

Authors evaluated the Radialis organ-targeted PET camera performance for clinical imaging results. Authors also showed some useful measured results using proposed PET cameras. Literature search and background looks a little bit fine. There are small English grammar mistakes. Organization of the proposed paper looks fine. The manuscript is very useful to be published Therefore, the manuscript could be minor revision.

1. In Line 14, there are unnecessary space before Sensitivity. Please check others in entire manuscript.

2. No mark of Figure 2(B) in Line 93.

3. Figure 3 fonts are not clear to be seen.

4. In Figure 9, are there any saturation points in left and right Figure 9 ?

5. Please provide ref. (Total average system sensitivity is determined as~) with ref. (https://iopscience.iop.org/article/10.1088/1361-6560/ab6579/meta)

6. I am wondering there are any signal-to-noise ratio value from measured image data.

7. It would be more beneficial if authors show the future work in conclusion section.

Round 2

Reviewer 1 Report

This article can be accepted for publication in Sensors.